# Low Prevalence of HPV Related Oropharyngeal Carcinogenesis in Northern Sardinia

**DOI:** 10.3390/cancers14174205

**Published:** 2022-08-30

**Authors:** Francesco Bussu, Narcisa Muresu, Claudia Crescio, Roberto Gallus, Davide Rizzo, Andrea Cossu, Illari Sechi, Mariantonietta Fedeli, Antonio Cossu, Giovanni Delogu, Andrea Piana

**Affiliations:** 1Department of Medicine, Surgery and Pharmacy, University of Sassari-ENT Division, AOU Sassari, 07100 Sassari, Italy; 2Department of Humanities and Social Sciences, University of Sassari, 07100 Sassari, Italy; 3Otolaryngology Division, Azienda Ospedaliera Universitaria, 07100 Sassari, Italy; 4Otolaryngology, Mater Olbia Hospital, 07026 Olbia, Italy; 5Department of Medicine, Surgery and Pharmacy, University of Sassari, 07100 Sassari, Italy; 6Facoltà di Medicina e Chirurgia, Università Cattolica del Sacro Cuore, 00168 Rome, Italy

**Keywords:** North Sardinia, OPSCC, HPV-driven carcinogenesis, treatment deintensification, HPV prevalence, diagnostic methods, specificity, false positives

## Abstract

**Simple Summary:**

Oropharyngeal squamous cell carcinomas (OPSCCs) are the only head and neck malignancy with a clear increase in prevalence in Western countries, due to the HPV epidemics with an increasing proportion of HPV-related OPSCCs. Such figures, however, are extremely variable around the globe. The present report is the first to assess the prevalence of HPV-related OPSCC in Sardinia, a relatively isolated population in the West. The rate of HPV-driven OPSCC in such a population is close to that of less developed areas, with clear implications on epidemiology, prognosis, and reliability of methods for assessing HPV-related carcinogenesis. In fact, in the present setting, the specificity of p16 IHC alone in diagnosing HPV-related carcinogenesis is only 75% with a 25% false positive rate.

**Abstract:**

HPV infection is a clear etiopathogenetic factor in oropharyngeal carcinogenesis and is associated with a markedly better prognosis than in smoking- and alcohol-associated cases, as specified by AJCC classification. The aim of the present work is to evaluate the prevalence of HPV-induced OPSCC in an insular area in the Mediterranean and to assess the reliability of p16 IHC (immunohistochemistry) alone, as accepted by AJCC, in the diagnosis of HPV-driven carcinogenesis in such a setting. All patients with OPSCC consecutively managed by the referral center in North Sardinia of head and neck tumor board of AOU Sassari, were recruited. Diagnosis of HPV-related OPCSS was carried out combining p16 IHC and DNA testing on FFPE samples and compared with the results of p16 IHC alone. Roughly 14% (9/62) of cases were positive for HPV-DNA and p16 IHC. Three more cases showed overexpression of p16, which has a 100% sensitivity, but only 75% specificity as standalone method for diagnosing HPV-driven carcinogenesis. The Cohen’s kappa coefficient of p16 IHC alone is 0.83 (excellent). However, if HPV-driven carcinogenesis diagnosed by p16 IHC alone was considered the criterion for treatment deintensification, 25% of p16 positive cases would have been wrongly submitted to deintensified treatment for tumors as aggressive as a p16 negative OPSCC. The currently accepted standard by AJCC (p16 IHC alone) harbors a high rate of false positive results, which appears risky for recommending treatment deintensification, and for this aim, in areas with a low prevalence of HPV-related OPSCC, it should be confirmed with HPV nucleic acid detection.

## 1. Introduction

In recent decades, a dramatic increase in oropharyngeal squamous cell carcinoma (OPSCC) incidence has been observed in Western countries. This burden of OPSCC appears to be related to the human papillomavirus (HPV) epidemic [1,2], which is now responsible for 60–80% of cases in the United States [3,4,5,6,7,8].

HPV-driven OPSCC has been clearly demonstrated to be a different entity compared to the traditional smoking-/alcohol-associated OPSCC, as has been acknowledged in the eighth American Joint Committee on Cancer (AJCC) staging system edition since 2016 [9]. A large amount of evidence confirms markedly better oncological outcomes of HPV-related OPSCC in comparison with HPV-negative counterpart [6,7,10], but a long-term morbidity with deterioration of quality of life (QoL) is often present in survivors [3,4]. The impact on QoL of aggressive treatments on a generally very responsive malignancy (HPV-related OPSCC) led to an intense debate about treatment de-intensification, with the aim to reduce long-term morbidity [5,6].

The global pooled prevalence of HPV-driven OPSCC has been estimated at ~45%, however, most data describe a huge geographic heterogeneity of such figure, with rates of HPV-induced OPSCC approaching 80% in the US and other Anglosaxon Countries (as New Zealand), as well as Scandinavia, and markedly lower rates in other countries and areas of the world: lower than 20% in highly populated countries as India and Brazil, and in some areas of China [3,4,5,6,7]. In general, available evidence suggests higher proportions of HPV-induced OPSCC in highly developed countries, in North America and Europe, and lower rates in Asia and Africa (accounting for the majority of world population).

This may have huge implications on overall prevalence of OPSCC, but also on the prognosis and on the clinical approach to the disease in different areas of the world.

The aim of the present work is to evaluate the prevalence of HPV-induced OPSCC in a small, relatively isolated insular area in the Mediterranean, by a validated assay combining p16 immunohistochemistry (IHC) and DNA testing [8,10]. A secondary endpoint is the assessment of the reliability of p16 IHC alone, as accepted by AJCC [9], in the diagnosis of HPV-driven carcinogenesis in such a setting.

## 2. Materials and Methods

Patients with a diagnosis of OPSCC, managed at the University Hospital of Sassari (Sardinia, Italy), between 2017 and 2021, were enrolled. TNM classification and staging of tumors according to AJCC (8th ed. 2017) [9] were assigned to each case based upon histopathological findings on the biopsy, imaging findings, and molecular analyses during multidisciplinary meetings of the institutional head and neck tumor board.

Personal (i.e., age, sex), clinical (i.e., anatomical site, stage, grade, HPV-DNA, p16-IHC), and follow-up data (i.e., outcome) were prospectively collected in an ad-hoc electronic form.

All the available data were analyzed in order to assess the prevalence of HPV-driven OPSCC and the histopathological and molecular-associated findings.

According to the algorithm described by VUMC group [8,10,11], we considered as HPV-driven only OPSCC with positive testing both for p16 IHC and GP5+/6+ DNA Real-time PCR.

The patients were managed and parameters collected following the current standard described by the main international guidelines [12]; therefore, data were analyzed with an observational retrospective design, and in this case a mandatory ethical approval was not requested by the Italian law (GU No. 76 31 March 2008).

### 2.1. Histopathological Analysis

Upon diagnosis of OPSCC, p16 IHC and GP5+/GP6+ were performed on formalin-fixed paraffin-embedded (FFPE) archives of the local Pathological Anatomy Unit. A total of 10 consecutive sections of 3 µm were cut and used for histological and molecular analyses. P16 expression was assessed using the CINtec Histology kit (Ventana Medical Systems, Inc., Tucson, AZ, USA), a qualitative immunohistochemistry test using mouse monoclonal anti-p16 antibody, following the manufacturer’s instruction. Samples were considered positive if a diffuse, continuous staining of basal and parabasal cell layers of the oropharyngeal squamous epithelium (>75%) was observed in a sequence of proliferating cells. Otherwise, negative status was defined as either focal p16 staining (i.e., staining of isolated cells or small cell clusters) or no p16 staining [13].

### 2.2. Molecular Analyses

After the deparaffinization step, the nucleic acid extraction was performed using the commercial AllPrep DNA/RNA FFPE kit (QIAGEN, Germany), following the manufacturer’s instruction [14]. DNA and RNA extracts were stored at −20 °C and analyzed within 48 h.

HPV-DNA detection was performed using the generic primers GP5+/GP6+ (forward 5′-TTTGTTACTGTGGTAGATACTAC-3′, reverse 5′-GAAAAATAAACTGTAAATCATATTC-3′), with the following thermal cycling conditions: 2 min at 95 °C, 40 cycles of 30 s at 95 °C, 30 s at 48 °C and 30 s at 72 °C, followed by 5 min at 72 °C. Real-time PCR was performed on CFX96 Real-time PCR system (BioRad). CCR5 housekeeping gene was used as an internal control of amplification.

HPV-positive samples were tested for the detection of E6-mRNA by RT-PCR. The retro-transcription of RNA was performed by the commercial kit Quantitect Reverse Transcription Kit (QIAGEN). The amplification step was performed using the protocol designed by Cocuzza et al., which allows the identification of E6 mRNA of HPV-16, -18, -31, -33, -35, -45, -58 and U1 gene, as internal control, as previously reported [15,16]. Qualitative real-time PCR was carried out using SYBR Green JumpStart, in a 25 µL reaction, with the following protocol: 94 °C for 2 min, followed by 40 cycles of 94 °C for 15 s and 60 °C for 1 min. HPV-16 and -18 positive cervical specimens and nuclease-free water were used as positive and negative controls, respectively.

### 2.3. Statistical Analysis

Statistical analysis was performed using the JMP in software, release 7.0.1, from the SAS Institute (Cary, NC, USA).

The correlations between p16 IHC and HPV status, and generally among the other categorical variables, were evaluated using Χ^2^ tests.

The correlation between detection methods was also determined by calculating the kappa value, a measure that takes a value of zero if there is no correlation between the samples. To score the significance, the kappa values were ranked. Values less than 0.4, between 0.4 and 0.75, or higher than 0.75 represent poor, fair-to-good, or excellent correlations, respectively.

## 3. Results

Sixty-two OPSCC patients were observed in the period under analysis. Descriptive statistics of demographic, molecular, and clinico-oncological characteristics are reported in Table 1.

Nine out of 62 cases (14.5%) turned out to be HPV-driven according to the previously defined criteria (p16 staining at IHC and hr-DNA amplification). All positive cases were confirmed at RT-PCR, six (66.7%) were associated with HPV16, one (11.1%) to HPV18, and two (22.2%) to HPV35.

The analyses for the detection of E6-mRNA, carried out only for HPV-16 and -18 positive samples, confirmed the presence of the previously diagnosed genotype and the expression of oncogenic genes.

In the present series, 12/62 cases were p16 positive, 3 of them (25%) tested negative for HPV DNA. In turn, all samples positive for hr-HPV DNA were also p16 positive. Sensitivity of p16 IHC alone in detecting HPV-induced carcinogenesis is therefore 100%, and specificity is 75%. It also means that 3/53 (5.66%) HPV-negative cases stained positive for p16.

In our previous works [17,18], the kappa coefficient to evaluate the performance of p16 as a surrogate marker resulted to be fair (between 0.6 and 0.7). In the present series, the value of kappa describing the reliability of p16 immunohistochemistry in assessing HPV-related carcinogenesis is 0.83 (excellent).

Disease-specific survival (DSS) in the present whole series is 68% at 3 years, HPV-driven carcinogenesis is associated with longer DSS (100% vs. 64% at 3 years), and statistical significance is lacking because of the small number of positive cases. If HPV-driven carcinogenesis were assessed by p16IHC alone, the possibility to predict DSS would have been clearly compromised, as shown in Figure 1.

## 4. Discussion

The present data demonstrate a prevalence of HPV-induced carcinogenesis among OPSCC in North Sardinia of 14.5%. It is useful information for a number of reasons. This is the first estimate, to our knowledge, of such a figure in the Sardinian population, and it is obtained using a widely validated, reliable, and conservative assay (combining p16 IHC and HPV nucleic acid detection) [8,19]. Sardinia is one of the less industrialized areas in Italy and is characterized by a certain isolation also for being an island, with specific genetic features of the inhabiting species and also of human population [20]. Even if definitive conclusions cannot be drawn because of the small numbers, the relatively high frequency of HPV35, an “uncommon”, but already described in OPSCC [21,22,23] high-risk HPV genotype, can be noted.

More interestingly, 14.5% is one of the lowest figures of HPV prevalence in OPSCC reported in Italy [17,18,24,25,26,27]. In fact, despite the wide variability, the reported rate of HPV-driven OPSCC in Italy in the last decade has always been above 30% [28,29]. This confirms a trend towards lower HPV carcinogenesis rates in the oropharynx in less developed areas, also within the West. The portion of HPV-driven OPSCCs in the present series is definitely closer to countries such as India and Brazil than to Anglosaxon and Scandinavian countries.

In general, when dealing with HPV-driven oropharyngeal carcinogenesis, a recurrent bias of clinical studies is the diagnostic tool [30]. Even if all authors agree that the gold standard for diagnosing HPV-driven carcinogenesis is E6 and E7 mRNA detection [8,30,31,32], the vast majority of studies use different assays, usually p16 IHC, a surrogate marker, which is considered adequate by the AJCC [9]. Not surprisingly, the low prevalence of HPV-induced carcinogenesis in the current series is associated with a high rate of false positive cases when HPV-induced carcinogenesis is diagnosed by p16 IHC alone (25%). In fact, the specificity of p16 IHC as a surrogate marker of HPV-induced carcinogenesis is extremely variable in the literature and is proportional to the proportion of OPSCCs induced by HPV [11,19,33,34,35]. In turn, the issue of false positives in HPV testing is extremely sensitive in head and neck oncology. In fact, HPV-driven OPSCCs being markedly more sensitive to treatments and characterized by a markedly better prognosis, one of the main and most discussed perspectives in oropharyngeal oncology remains treatment deintensification [36,37]. It has been demonstrated that p16-positive HPV-negative OPSCCs are characterized by the same prognosis as p16-negative cases [10,11]; therefore, if the indication to de-intensified treatments would be given on the basis of p16 IHC alone, the former cases would be probably undertreated and treatment de-escalation would probably result in being detrimental on prognosis because of a higher risk of locoregional relapse [38]. Any standardization of de-intensification protocols in the management of HPV+OPSCC will first require a consensus on highly specific diagnostic assays [19,30].

A cut-off should be established for the specificity, below which p16 IHC alone should not be considered sufficient anymore to diagnose HPV-driven oropharyngeal carcinogenesis. It means that the current TNM classification for the oropharynx, which considers p16 immunohistochemistry sufficient for diagnosing HPV-related OPSCC, loses reliability among populations with lower rates of HPV-driven cancers, which is probably the majority of the world population, including Sardinia.

Notably, if the HPV-related OPSCC rate is extremely variable, in the series where p16 IHC and nucleic acid detection are both performed, the false positive rate (FPR) of p16 IHC among HPV-negative cases is between 5 and 20% [11,18,35,39,40,41,42,43,44,45,46], and is pretty constant for a given assay even in different populations (such as Dutch and Sardinians) (ISH, DNA detection without amplification, PCR) (some examples are shown in Table 2). When the current standard for the diagnosis of HPV-driven carcinogenesis is used (mRNA detection or sequential p16IHC and HPV-DNA PCR), the rate of HPV-negative OPSCC expressing p16 is notably always about 5.5% [11,44,45,46]. It is probably due to the fact that a constant proportion of HPV-negative OPSCCs overexpresses p16. It means that the specificity of p16 IHC can be easily predicted based upon the prevalence of HPV-driven oropharyngeal carcinogenesis, and when it will be higher than a previously established cut-off, integration through hr-HPV nucleic acid detection, as recommended by Dutch authors [8,11,47], should be considered mandatory before defining an OPSCC as HPV-driven.

## 5. Conclusions

The present report assesses the prevalence of HPV-related OPSCC in Northern Sardinia, a relatively isolated population showing a different epidemiology from other Italian regions and European trends, closer to less developed areas. This scenario leads to clear implications on prognosis and reliability of methods for assessing HPV-related carcinogenesis. In fact, in the presence of low prevalence of HPV-induced tumors, a high rate of false-positive cases diagnosed by p16 IHC is systematically reported. For this reason, p16 alone cannot be considered an adequate surrogate marker for HPV detection. We believe that in such cases, integration through hr-HPV nucleic acid detection should be mandatory.

## Figures and Tables

**Figure 1 cancers-14-04205-f001:**
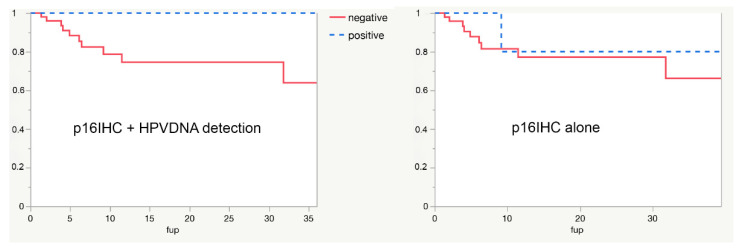
Survival curves in HPV-positive versus HPV-negative cases in the present series. When HPV-driven carcinogenesis is assessed by p16IHC alone (right panel), the power to predict survival is clearly compromised.

**Table 1 cancers-14-04205-t001:** Descriptive statistics.

*Male, n (%)*		54 (87.1)
*Mean age (SD), years*		64 (10.2)
*Anatomic site, n/N (%)*	*Tonsil/Lateral wall*	22/62 (35.5)
	*Base of Tongue (BOT)*	19/62 (30.7)
	*Posterior wall*	11/62 (17.7)
	*Soft palate*	10/62 (16.1)
*Tumor Stage, n/N (%)*	*I*	6/62 (9.7)
	*II*	10/62 (16.1)
	*III*	6/62 (9.7)
	*IV*	40/62 (64.5)
*Histopathological grading (AJCC), n/N (%)*	*not assessed*	24/62 (38.7)
	*G1*	0/62 (0.0)
	*G2*	20/62 (32.3)
	*G3*	18/62 (29)
*HPV-DNA positivity, n/N (%)*	9/62 (14.5)	
	*HPV-16*	6/9 (66.7)
	*HPV-18*	1/9 (11.1)
	*HPV-35*	2/9 (22.2)
*p-16 immunohistochemistry positivity, n/N (%)*	12/62 (19.4)	

**Table 2 cancers-14-04205-t002:** Reliability of p16 IHC in different series of OPSCCs. Notably, the FPR is pretty constant for a certain assay even in very different populations (such as Dutch, Germans, Czechs, and Sardinians), while specificity decreases with the HPV-driven rate.

Population	Rate of p16 Positive among HPV-Negative OPSCCs (FPR)	Rate of HPV-Induced OPSCC in the Population	Proportion of HPV-/p16+ OPSCC in the Population	Probability That Positive p16 Is HPV-(1-Specificity)
**Nauta (Holland)**	**5.6%**	28.2%	4%	12.3%
**Bussu (North Sardinia)**	**5.7%**	14.5%	4.8%	25%
**Bussu (Central Italy)**	**20.6%**	32%	14%	30.4%
**Saito (Japan)**	**9.8%**	32%	6.7%	17.2%
**Benzerdjeb (France)**	**13.6%**	46.4%	7.2%	13.6%
**Schache (UK)**	**11.3%**	36.1%	7.2%	18%
**Ang * (US)**	**18.8%**	63.8%	6.8%	10.3%
**Rotnaglova (Czech Republic)**	**5.3%**	60%	2.2%	3.7%
**Linge (Germany)**	**9.6%**	21.7%	7.4%	25.6%
**Prigge (Germany)**	**5.7%**	17.2%	4.7%	21.4%
**Carolina Oliva (Chile)**	**26.3%**	60.4%	10.4%	17.9%
**Galo Méndez-Matías (Mexico)**	**20%**	39.2%	12.3%	23.7%

* HPV detected through FISH, with notorious sensitivity issues, with a relevant rate of false negative cases.

## Data Availability

The data is available when it is requested by motivated reasons.

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
