# Peer review of "Low Prevalence of HPV Related Oropharyngeal Carcinogenesis in Northern Sardinia"

_cancers, 2022, doi:10.3390/cancers14174205_

Round 1
Reviewer 1 Report
Comments in general:
- Relevant to study and report on the HPV-attributable fraction of Sardinia
- The study was executed correctly
- The message is very important
Feedback on Abstract:
- Line 25: the abbreviation p16-IHC is used for the first time in line 25, but is only explained (IHC = immunohistochemistry) in line 29. Please explain the p16-IHC abbreviation in line 25 instead of line 29.
- Line 36: “The currently accepted standard by AJCC (p16 IHC alone) harbors a high 36 rate of false positive results, which is unacceptable for recommending treatment deintensification, 37 in areas with a low prevalence of HPV-related OPSCC” à I find this statement too strong. The AJCC agreed on p16-IHC alone as surrogate marker for HPV, since it is cheap and easy to perform, and every hospital/lab in the world is able to perform p16-IHC. On the contrary, HPV DNA testing is more expensive and some less developed countries might not be able to perform HPV DNA testing. I think the message should be that, in case HPV DNA testing can be performed, it should, and not only for correct staging, but especially when considering patients for treatment de-intensification.
Feedback on Introduction:
- Line 44: “In recent decades, we assisted in a dramatic increase” à not the correct term, alternative: “we witnessed”?
- Line 44: “oropharyngeal squamous cell carcinoma (OPSCC) incidence in western countries” à Western with capital W.
- Line 48: “HPV driven OPSCC has been clearly demonstrated to be a different entity com-48 pared to the traditional smoking/alcohol-associated OPSCC, as it is acknowledged in the 49 8th American Joint Committee on Cancer (AJCC) staging system edition since 2016” à please provide appropriate references.
Feedback on Material and Methods:
- Line 77: “based upon histopathological findings on the biopsy, imaging finding and molecular” à findings
- Line 80: “Personal (i.e.; age, sex), clinical (i.e.; anatomical site, stage, grade, HPV-DNA, p-16 IHC” à p16-IHC
- Line 81: “IHC, outcome), and follow up data have been prospectively collected in an ad hoc elec-81 tronic form” à doesn’t outcome belong to follow-up data instead of clinical data?
- Line 99: “Samples were considered positive if a strong positive signal were observed in a sequence of pro-100 liferating cells [14]” à what cut-off value did you use for p16-positivity? Is not stated in the reference. Normally it’s >70% moderate to strong nuclear and cytoplasmic immunoreactivity. Please specify.
Feedback on results:
- I miss information on survival! In the Material and Methods it’s stated that outcome and follow-up data were collected, but they’re not mentioned in the results. Could you please provide at least 3- or 5-year overall survival for p16-negatieve, p16-positive/HPV DNA-positive and possibly also for p16-positive/HPV DNA-negative, although the patient groups are small? Because statements on prognosis are made in the abstract and discussion, but are not specified for your cohort.
- Line 137: In table 1 you mention histopathological grading, but I only see G2 and G3. What about G1?
- Line 137: table 1: HPV DNA positivity, n (%) à should be n/N (%)
- Line 137: table 1: p16-immunohistochemistry positivity, n (%) à should be n/N (%)
Reviewer 2 Report
The present report assesses the prevalence of HPV related OPSCC in Northern Sardinia and compares it to p16 results. The authors recommend that p16 alone cannot be considered an adequate surrogate marker for HPV detection.
Minor comments
Line 30 - ~14% (9/62) cases were positive to HPV-DNA - Change 'to' to 'for'
Line 31- 3 more cases resulted to overexpress p16, - Change to 5 more cases showed overexpression of p16
Line 44 - This statement is unclear - we assisted in a dramatic increase
Inconsistency in reporting proportions for example - 14.5% and 14,5%. Please make that consistent.
